# Two Birds with One Stone: Preparation of 4, 4-Diaminodiphenylmethane Functionalized GO@SiO_2_ with Mechanical Reinforcement and UV Shielding Properties and Its Application in Thermoplastic Polyurethane

**DOI:** 10.3390/polym13234220

**Published:** 2021-12-01

**Authors:** Guoxin Ding, Hongxu Tai, Chuanxin Chen, Chenfeng Sun, Zhongfeng Tang

**Affiliations:** 1School of Materials Science and Engineering, Anhui University of Science and Technology, Huainan 232001, China; thx0315@163.com (H.T.); CCX0720@foxmail.com (C.C.); The_Shadower@163.com (C.S.); 2Shanghai Institute of Applied Physics, Chinese Academy of Sciences, Shanghai 201800, China

**Keywords:** graphene oxide, silicon dioxide, mechanical properties, thermoplastic polyurethane, ultraviolet

## Abstract

This study prepared 4,4-diaminodiphenylmethane (DDM)-functionalized graphene oxide (GO)@silica dioxide (SiO_2_) nano-composites through amidation reaction and low-temperature precipitation. The resulting modified GO, that was DDM−GO@SiO_2_. The study found that DDM−GO@SiO_2_ showed good dispersion and compatibility with thermoplastic polyurethane (TPU) substrates. Compared with pure TPU, the tensile strength of the TPU composites increased by 41% to 94.6 MPa at only 0.5 wt% DDM−GO@SiO_2_. In addition, even when a small amount of DDM−GO@SiO_2_ was added, the UV absorption of TPU composites increased significantly, TPU composites can achieve a UV shielding efficiency of 95.21% in the UV-A region. These results show that this type of material holds great promise for the preparation of functional coatings and film materials with high strength and weather resistance.

## 1. Introduction

Thermoplastic polyurethane (TPU) is a multi-segment copolymer consisting of hard and soft segments [1]. It is also an important elastomeric polymer. Given its abundant sources and good processing properties, TPU is widely used in packaging materials, functional coatings, and other applications [2,3,4]. However, the applications of TPU are limited by its low mechanical strength and poor weather resistance [5,6,7]. In addition, TPU is prone to automatic oxidation and yellowing under UV light, making this polymer brittle and yellow [8,9]. Thus, many functional materials have been used to prepare TPU-based composites [10,11]. Two-dimensional carbon materials, particularly graphene oxide (GO), have been widely used for the production of high-performance polymers because of their unique mechanical and optical properties [12,13]. The key issues in GO/polymer nanocomposites are how to address dispersion and interfacial forces [14]. Chemical modification is an effective way of solving both problems [15,16,17]. Ren et al. [18] used ethylenediamine to prepare cross-linked GO and found that ethylenediamine filled into TPU improves the dispersion of GO while significantly increasing the tensile strength and elongation at break. Halima et al. [19] used acetamide-modified GO, which is homogeneously dispersed into PU as a filler, to improve the anti-corrosion properties of PU. The dispersion of GO in polymers can also be improved by modifying it with nanomaterials [20]. The in situ growth of silica (SiO_2_) on the surface of GO improves dispersion and reduces interfacial forces; meanwhile, the modification of SiO_2_ imparts a highly porous surface structure to GO, which provides non-covalent attachment points and improves the physical entanglement between GO and TPU [21]. Kim et al. [22] attached nano-SiO_2_ onto the surface of GO by using the sol-gel method and found that silica modification enhances the polarity of the surface, which generates a chemical affinity between the SiO_2_/GO nanocomposite and TPU. Yu et al. [21] obtained graphene/SiO_2_ composites by in situ preparation and introduced the hybrid nanoparticles into epoxy resins. They found that the unique 3D structure of graphene/SiO_2_ increases the interfacial adhesion between the filler and the matrix and significantly improves the mechanical properties of the epoxy resins. SiO_2_/GO nanocomposites not only improve the mechanical properties of the polymer but also protect against UV light. The conjugated groups and aromatic structures on the surface of GO provide good absorption of UV light [23], whereas those of SiO_2_ can reflect UV light [24] and can be added to polymers to act as a barrier and reduce damage caused by UV light to the polymer matrix.

This study proposed a simple and environment-friendly method. DDM−GO@SiO_2_ nanocomposites were obtained by the hydrophobic modification of GO using 4,4-diaminodiphenylmethane (DDM) and in situ chemical growth of SiO_2_ nanocomposites on DDM−GO by using potassium silicate water glass. DDM−GO@SiO_2_ was dispersed into TPU by solution blending. DDM−GO@SiO_2_ was successfully characterized using scanning electron microscopy (SEM), infrared (IR) spectroscopy, X-ray diffraction (XRD), Raman spectroscopy, X-ray photoelectron spectroscopy (XPS), and other methods. The effects of different additions of DDM−GO@SiO_2_ on the mechanical properties and UV shielding properties of TPU were also focused on.

## 2. Materials and Methods

### 2.1. Materials

GO was supplied by Tan Feng Tech., Inc, Jiangsu, China. DDM, ammonium hydroxide, N, *N*-dimethylformamide (DMF), ethyl alcohol, and sodium dodecyl benzene sulfonate (SDBS) were obtained from Sinopharm Chemical Reagent Co., Ltd., Shanghai, China. Hydrochloric acid with an HCl content of 37.5% was purchased from Jia Ni Chemical Co., Ltd., Jiangsu, China. These reagents are of analytical grade. TPU (WHT-1195) was acquired from Wan Hua Chemical Group Co., Ltd., Hubei, China. Potassium silicate was supplied by Usolf Chemical Tech., Inc, Shandong, China. Deionized water was homemade and then used for all experimental procedures.

### 2.2. Preparation of DDM−GO@SiO_2_ Particle

1.5 g of DDM was added at 95 °C to 300 mL of deionised water sonicated with 0.5 g of GO. The mixture was magnetically stirred at 95 °C for 10 h and then DDM−GO was obtained by vacuum filtration of the mixture several times using deionized water and freeze-dried.

3.0 g potassium silicate was added to 50 mL of deionised water sonicated with 0.5 g DDM−GO and 0.2 g SDBS. The pH of the mixture was adjusted to 7 by hydrochloric acid in an ice water bath and maintained for 12 h. Lastly, DDM−GO@SiO_2_ composites were obtained by vacuum filtration of the mixture several times using deionized water and freeze-dried.

### 2.3. Preparation of the DDM−GO@SiO_2_/TPU Composites

DDM−GO@SiO_2_ composites were added to 100 mL of DMF, ultrasonically dispersed for 20 min, and then heated to 80 °C. TPU (10.0 g) was dissolved into the dispersion, stirred at high speed for 4 h, placed in a mold, and de-bubbled in a vacuum drying oven at 80 °C. Then, the DMF was dried out to prepare DDM−GO@SiO_2_/TPU composites. DDM−GO@SiO_2_ nanocomposites were filled with 0, 0.1, 0.3, 0.5, 1, and 2 wt% of the total TPU and denoted as S0, S1, S2, S3, S4, and S5, respectively. Figure 1 showed the preparation procedure of DDM−GO@SiO_2_/TPU composites.

### 2.4. Characterization

Characteristic peaks of the material were analyzed via Fourier-transform infrared (FT-IR) spectroscopy (AVATAR370, Thermo Nicolet, MA, USA). The crystal structure of the material was characterized using XRD (Smartlab SE, Japanese Science, Tokyo, Japan) with a scanning diffraction angle of 5°–80° and a step size of 5°/min. The chemical composition of the material was determined with XPS (ESCALAB 250Xi, Thermo Fischer, MA, USA). Bond and structural defects of the material were analyzed using Raman spectroscopy (In Via, Renishaw, Gloucestershire, UK) at a wavelength of 532 nm. The microstructure of the material was examined via SEM (Gemini 300, ZEISS, Oberkochen, Germany) operating at 10 kV, and local elemental analysis was performed via energy dispersive spectroscopy (EDS, Xplore, Oxford, UK) at 20 kV. Cross-sectional scanning of the TPU composites was brittle broke in liquid nitrogen. The specimens were subjected to tensile tests using GB/T 528-2009. The testing apparatus was a universal testing machine (WDW-50, Kaiqiangli Testing Apparatus Co., Ltd., Shenzhen, China). Absorbance and transmittance of the materials were tested on a UV-VIS-NIR spectrophotometer (Lambda950, PerkinElmer, MA, USA).

## 3. Results and Discussion

### 3.1. DDM−GO@SiO_2_ Particles Structure

Figure 1a displayed the FT-IR spectra of GO, DDM−GO, and DDM−GO@SiO_2_ particles. The FT-IR spectrum of GO demonstrates O–H vibration, C=O vibration, C=C stretching vibration, C–H vibration, and C–O stretching vibration peaks at 3406, 1723, 1626, 2923, and 1069 cm^−1^, respectively [25]. The FT-IR spectrum of DDM−GO shows that the peak at 1038 cm^−1^ is the stretching vibration of C–N [26]. The decrease in intensity of the absorption peak associated with the carboxyl group is due to a decrease in the number of oxygen-containing groups on the GO surface. Amination reaction result in the red shift of C=O at 1723–1630 cm^−1^ because of p–π conjugation [27]. These results indicate that GO is successfully reduced via DDM. As shown in Figure 1b, the Si–OH and Si–O–Si peaks of DDM−GO@SiO_2_ at 800 and 477 cm^−1^, respectively, are consistent with the characteristic peaks of SiO_2_ [28]. The Si–O–C stretching vibration peak appears at 1054 cm^−1^, and O–H presents a lower peak intensity compared with DDM−GO possibly because of the reaction of the hydroxyl group on the surface of DDM−GO with Si–OH on the surface of SiO_2_ to generate Si–O–C via dehydration [29]. Hence, the reduction of the hydroxyl group on the surface of DDM−GO indicates the successful grafting of SiO_2_ on the surface of DDM−GO.

XPS tests were performed on DDM−GO@SiO_2_ to assess whether or not the modification was successful. The total XPS spectrum of DDM−GO@SiO_2_ in Figure 2a presents characteristic peaks corresponding to Si2p, C1s, N1s, and O1s. N element is derived from the modifier DDM, and Si element is derived from the SiO_2_ graft on the DDM−GO surface. Figure 2b shows the presence of N element in three forms. Fitted peaks of N1s at 398.6, 399.2, and 399.8 eV correspond to Pyridinic N, Pyrrolic N, and Graphitic N [30]. As shown in Figure 2c, fitted peaks at 104.2, 104.5, and 105.3 eV for Si2p correspond to Si–O–C, Si–O–Si, and Si–OH, respectively [31]. As shown in Figure 2d, fitted peaks at 284.1, 284.9, 286.1, 290.7and for C1s correspond to C–OH, C–N, C–COOH and C*, respectively [32].

XRD spectra of GO, DDM−GO and DDM−GO@SiO_2_ were illustrated in Figure 3a. GO presents two diffraction peaks at 2θ = 11.7° and 42.5°, which correspond to (001) and (100) crystal planes [33]. The comparison of XRD spectra of DDM−GO and GO show that the (001) crystalline plane disappears and a new diffraction peak appears at 2θ = 20.1°, indicating the successful reduction of GO [34].The XRD spectrum of DDM−GO@SiO_2_ shows a broad peak at 2θ = 22.7°, which corresponding to the (111) crystal plane of SiO_2_ nanoparticles. This result suggests that the SiO_2_ particles cover the layered GO sheets successfully.

Raman spectra of GO, DDM−GO, and DDM−GO@SiO_2_ show two distinctive peaks attributed to the G and D bands. The G band, which represents the E2g vibrational mode of first-order scattering and is used to characterize the sp2 bond structure of carbon, appeared at 1591 cm^−1^. The D band, which is a reflection of the disorder of the crystalline structure, appeared at 1347 cm^−1^ [35]. The ratio of D band/G band (I_D_/I_G_) intensity is an indicator of the extent of surface defects in carbon materials. The increment in D peak intensity implies the broken sp2 bonds and the presence of more sp3 bonds on the sheets. As shown in Figure 3b, the I_D_/I_G_ ratio was calculated to be 0.84 for GO, 0.94 for DDM−GO, and 1.15 for DDM−GO@SiO_2_. The higher I_D_/I_G_ value of DDM−GO than GO is due to the reaction of DDM with oxygen-containing groups on the GO surface, resulting in more surface defects after GO deoxidation. Meanwhile, the higher I_D_/I_G_ value of DDM−GO@SiO_2_ than DDM−GO indicates the increased disorder caused by the grafting of SiO_2_ onto DDM−GO. The results are similar to the study of Kou [36].

Microscopic morphologies of GO, DDM−GO, and DDM−GO@SiO_2_ were analyzed through SEM. Figure 4a,b show the smooth surface of GO sheet and the wrinkles on the reduced DDM−GO sheet. DDM as a reducing agent destroys the hexagonal carbon backbone of GO, introducing defects and loss of long-range ordering on the surface of the graphene sheet, causing it to wrinkle [37]. As shown in Figure 4c, the surface of DDM−GO@SiO_2_ is rougher than that of DDM−GO with many attached SiO_2_ particles. By contrast, the morphology of DDM−GO@SiO_2_ is similar to that of GO modified with a tetraethylsilicate mixture [38]. The EDS mapping analysis in Figure 4d is used to confirm the distribution of SiO_2_ on DDM−GO. Successful modification of GO by DDM was further demonstrated due to the presence of N elemental. It can be seen that Si element is evenly distributed in the system, which shows that SiO_2_ is well dispersed in DDM−GO.

### 3.2. Mechanical and UV-Shielding Performance Analysis

The stress–strain analysis was used to investigate the mechanical properties of TPU and the effect of DDM−GO@SiO_2_ content on the mechanical properties of the nanocomposites. Stress–strain curves and the corresponding tensile strength and elongation at break of the TPU composites with different DDM−GO@SiO_2_ fillings were shown in Figure 5a,b. Compared with that of pure TPU, the tensile strength of the TPU composites with 0.5 wt% DDM−GO@SiO_2_ filling content increased by 41% to 94.6 MPa. The significant increase in tensile strength can be attributed to the increased forces between DDM−GO@SiO_2_ and TPU. However, the tensile strength of the nanocomposites gradually decreased as the content of DDM−GO@SiO_2_ exceeded 0.5 wt% The elongation at break also decreased with increasing DDM−GO@SiO_2_ content. This phenomenon can be explained by the fact that the degree of deformation of the TPU molecular chains is limited at high DDM−GO@SiO_2_ loadings, leading to a reduction in the strain at break. Furthermore, a high amount of DDM−GO@SiO_2_ may lead to aggregation in the TPU matrix and induce the formation of defects, leading to a decline in mechanical properties [39]. The stretching vibration peak of N-H in the TPU matrix is very sensitive to changes in hydrogen bonding. Therefore, the hydrogen bonding changes between DDM−GO@SiO_2_ and TPU were characterised by FT-IR tests. As shown in Figure 5c, with the addition of DDM−GO@SiO_2_, the N-H stretching vibration peak of the TPU composite blue shifts from 3331 cm^−1^ to 3333 cm^−1^, which is attributed to the addition of the filler breaking the hydrogen bond within the TPU and forming a stronger hydrogen bond between the filler and TPU matrix. The hydrogen bond between DDM−GO@SiO_2_ and TPU is shown schematically in Figure 5f. However, when the filler content is further increased, the peak is red shifted, which is attributed to agglomeration of filler affecting the formation of hydrogen bonds. In addition, as shown in Figure 5d, the stretching vibration peaks at 1732 cm^−1^ and 1703 cm^−1^ correspond to “free” C=O and bonded C=O in TPU, respectively. For every N-H bond created between DDM−GO@SiO_2_ and TPU, a new “free” C=O is formed [40]. A Gaussian fit is made to the curve in Figure 5d to investigate the variation in C=O. As shown in Figure 5e, the free C=O peak area/C=O peak area values of the TPU composites are higher than pure TPU [41,42], indicating a strong interaction between DDM−GO@SiO_2_ and TPU, the trend also corresponds to the change in mechanical properties.

The transmittance and absorbance spectra of samples, including pure TPU and TPU nanocomposites with different DDM−GO@SiO_2_ contents from 0.1 wt% to 2 wt%, were obtained to analyze the UV-shielding performance of the DDM−GO@SiO_2_/TPU nanocomposites over the entire UV range. Figure 6a, b showed the absorbance and transmittance spectra of the samples in the wavelength range of 200–800 nm. The strong and wide UV absorption band at 321 nm is observed in pure TPU, accredited as n-π* leap absorption peak from C=O [43]. The UV absorption band at 233 nm is observed in the TPU composites, accredited as π-π* leap absorption peak from C=C. Notably, the C=O absorption peak of the TPU nanocomposite appears at 339 nm, showing a significant red shift along with greater absorption because of the enhanced interaction of DDM−GO@SiO_2_ and TPU [44]. The absorption peak of SiO_2_ is observed at a peak of 380 nm. The introduction of DDM−GO@SiO_2_ increases the absorbance of TPU within the entire UV region (200–400 nm), and the absorbance increases gradually with increasing content. As shown in Figure 6b, the transmittance of the TPU composites is reduced compared with that of pure TPU, especially in the UV-A and UV-B regions, where the transmittance decreases significantly with increasing DDM−GO@SiO_2_ content. This result is due to the reflection of UV light by SiO_2_ and the strong absorption of incident UV light by the GO sheet layer, thus protecting the internal structure of TPU. The relevant schematic is shown in Figure 6d. The blocking efficiency in the UV-A and B regions is summarized in Table 1 to quantify the UV shielding property of the DDM−GO@SiO_2_/TPU nanocomposites. The blocking efficiency at given UV ranges can be calculated using the following equations [45]:(1)$$\text{UV-A blocking }\left(\%\right)\text{ = 100 }-\;\frac{\int_{\text{320}}^{\text{400}}\text{T(λ)dλ}}{\int_{\text{320}}^{\text{400}}\text{dλ}}$$
(2)$$\text{UV-B blocking }\left(\%\right)\text{ = 100 }-\;\frac{\int_{\text{280}}^{\text{320}}\text{T(λ)dλ}}{\int_{\text{280}}^{\text{320}}\text{dλ}}$$

As shown in Figure 6c, the addition of DDM−GO@SiO_2_ improves the UV shielding efficiency in the UV-A and UV-B regions, especially in the UV-A region, which is the most serious for TPU UV aging. When the content of DDM−GO@SiO_2_ is 2 wt%, the shielding efficiency can reach 95.21% from 79.25% of pure TPU. The specific data are shown in Table 1.

### 3.3. Cross-Sectional Morphology of TPU Composites Analysis

The cross-sectional morphology of the polymer was characterized using SEM to investigate the dispersion and interfacial interaction of DDM−GO@SiO_2_ in TPU. Wavy and relatively smooth cross-sectional morphology is observed in pure TPU, as shown in Figure 7a. Similar fracture surfaces are also observed in the TPU composites (Figure 7b–f). In addition, the roughness at the fracture interface of the TPU composite become richer, which may be caused by crack bridging and pull-out DDM−GO@SiO_2_. As the loading of DDM−GO@SiO_2_ increased above 1 wt%, significant agglomeration and cracks occur in the cross section of the TPU composites (Figure 7e,f). Many cavities and agglomerates appear on the fracture surface with the increase in DDM−GO@SiO_2_ filling because of the agglomeration of the filler in TPU and the decrease in the interfacial bonding force. It is completely pulled out during the tensile process. The higher magnification SEM image and EDS spectra of the circular area of the 0.5 wt% DDM−GO@SiO_2_/TPU composite are shown in Figure 7g,h. The presence of Si and N elements can be clearly detected by EDS [46].

## 4. Conclusions

In this work, DDM was used to hydrophobically modify GO to improve dispersion in TPU. DDM−GO@SiO_2_ nanocomposites were obtained by depositing nano-SiO_2_ on the surface of DDM−GO by in situ growth. The introduction of nano-SiO_2_ provides non-covalent attachment points and further improves DDM−GO physical entanglement with TPU. Meanwhile, DDM−GO and SiO_2_ have a synergistic effect in reinforcing and shielding UV light. TPU composites with different DDM−GO@SiO_2_ contents were prepared by solution blending. Compared with that of pure TPU, the tensile strength of the TPU composites with a filling amount of 0.5 wt% increased by 41%, reaching 94.6 MPa. Benefiting from the absorption of UV light by GO and reflection by SiO_2_, in the UV-A region, when the DDM−GO@SiO_2_ content is only 2 wt%, the transmission rate is significantly reduced, and its shielding efficiency can reach 95.21% from 79.25% of pure TPU. In addition, the DDM−GO@SiO_2_/TPU composites have the advantages of a simple preparation process and low cost. This type of material holds great promise for functional coatings and high-performance thin film materials.

## Data Availability

This study did not report any data.

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
