# Peer review of "Two Birds with One Stone: Preparation of 4, 4-Diaminodiphenylmethane Functionalized GO@SiO2 with Mechanical Reinforcement and UV Shielding Properties and Its Application in Thermoplastic Polyurethane"

_polymers, 2021, doi:10.3390/polym13234220_

Round 1

Reviewer 1 Report

Dear Editor,

The manuscript (polymers-1486028) reports development of DDM-GO@SiO2 incorporated thermoplastic polyurethane (TPU) as a UV-shielding material. The study is interesting, the manuscript reads well, and the graphs, images and schematics have been presented well. However, there are several major concerns that need to be taken care of:

1) The title contains several abbreviations that should be written in full.

2) In general, several abbreviations in abstract are defined after the first appearance, such as DDM and TPU.

3) Abstract is written non-concisely, for instance, line 10, what kind of chemical reaction, line 12, what similar structure, line 18, what kind of coating and film,

4) Page 5, line 143; what amorphous crystallinity means?

5) The emphasis of the manuscript is on the filler rather than the composite as whole. 

6) Figure 4, can the authors mark each phase, GO and SiO2, in the respective SEM images?

7) Page 6, line 180-181; this claim (uniform dispersion) should be validated. It is more related to intermolecular bonding of TPU and DDM-GO@SiO2 that should be characterized via FTIR.

8) Page 6, line 186; aggregation represents poor interaction between the polymer matrix and the filler, thus implies failure of material selection and design.

9) Table 1 reports the values shown in Fig. 5b thus is unnecessary.

Reviewer 2 Report

Ref.comments to the paper titled as “Two birds with one stone: preparation of DDM-GO@SiO2 with mechanical reinforcement and UV shielding properties and its application in TPU”  written by the authors: Guoxin Ding , Hongxu Tai, Chuanxin Chen, Chenfeng Sun, Zhongfeng Tang

It is well known, that the development of graphene-based structures permits to find the novel way to apply the modified materials in the general optoelectronics, laser physics, biomedicine, etc. area.  Thus, from this point of view the current paper is modern and actual.

For the first, it is remarkable that the authors have made a complicated approach with including the different aspects in this direction. The authors have analyzed 43 scientific publications in the analogous area, including the manuscripts published on last 5 years. This indicates the knowledge of the problem and real practical ways to solve it.

The section on materials and methods is quite comparable with our knowledge in the field of technological processes for the synthesis of new materials and does not contradict with the physico-chemical trends and knowledge. Cheme1. - Preparation procedure of DDM-GO@SiO2/TPU composites - is supported the process with good advantage.

Results and discussion sections are well explained the basic data. Spectral analysis and XPS tests are shown the change in the transparency and in the intensity dependence on binding energy. They are included so many structural peaks and explained well. Raman shifts are coincided with the SEM-images presented. Mechanical properties are comparatively shown in Table 1. -Mechanical properties data of DDM-GO@SiO2/TPU nanocomposites with different filling amount – that can be useful for the practical application as well. The strong absorption of an incident ultra violet light by the graphene oxides sheet layer taken into account in order to explain the data shown in Fig.6.

Conclusion part is accumulated all basic results established in this paper.

The basic question to the authors is the following: What is the maximum intensity of your ultraviolet radiation that completely destroys the novel composite?

So, the article is good. In my local opinion, this paper can be published in the Journal with the corrections about the threshold of the UV intensity mentioned above.

Round 2

Reviewer 1 Report

Dear Editor,

Considering the applied revisions, I recommend the paper for publication.